# Molecular insights into intra-complex signal transmission during stressosome activation

Algirdas Miksys[1,8,10], Lifei Fu[1,10], M. Gregor Madej [1,10], Duarte N. Guerreiro [2], Susann Kaltwasser [3], Maria Conway[4], Sema Ejder [5], Astrid Bruckmann [6], Jon Marles-Wright [5], Richard J. Lewis[7,9], Conor O'Byrne[2], Jan Pané-Farré [4✉] & Christine Ziegler [1✉]

The stressosome is a pseudo-icosahedral megadalton bacterial stress-sensing protein complex consisting of several copies of two STAS-domain proteins, RsbR and RsbS, and the kinase RsbT. Upon perception of environmental stress multiple copies of RsbT are released from the surface of the stressosome. Free RsbT activates downstream proteins to elicit a global cellular response, such as the activation of the general stress response in Gram-positive bacteria. The molecular events triggering RsbT release from the stressosome surface remain poorly understood. Here we present the map of *Listeria innocua* RsbR1/RsbS complex at resolutions of 3.45 Å for the STAS domain core in icosahedral symmetry and of 3.87 Å for the STAS domain and N-terminal sensors in D2 symmetry, respectively. The structure reveals a conformational change in the STAS domain linked to phosphorylation in RsbR. Docking studies indicate that allosteric RsbT binding to the conformationally flexible N-terminal sensor domain of RsbR affects the affinity of RsbS towards RsbT. Our results bring to focus the molecular events within the stressosome complex and further our understanding of this ubiquitous signaling hub.

[1] Department of Biophysics II / Structural Biology, University of Regensburg, Regensburg 93053, Germany. [2] Bacterial Stress Response Group, Microbiology, School of Natural Sciences & Ryan Institute, National University of Ireland Galway, Galway H91 TK33, Ireland. [3] Department of Structural Biology, Max-Planck Institute of Biophysics, Frankfurt am Main 60438, Germany. [4] Center for Synthetic Microbiology (SYNMIKRO) and Department of Chemistry, Philipps-University Marburg, 35043 Marburg, Germany. [5] School of Natural and Environmental Sciences, Newcastle University, Newcastle upon Tyne NE1 7RU, UK. [6] Department of Biochemistry I, University of Regensburg, Regensburg 93053, Germany. [7] Newcastle University Biosciences Institute, Faculty of Medical Sciences, Newcastle University, Newcastle upon Tyne NE2 4HH, UK. [8] Present address: VU LSC-EMBL Partnership for Genome Editing Technologies, Life Sciences Center, Vilnius University, Vilnius, Lithuania. [9] Present address: The Royal Society for the Protection of Birds, The Lodge, Potton Road, Sandy SG19 2DL Bedfordshire, UK. [10] These authors contributed equally: Algirdas Miksys, Lifei Fu, M. Gregor Madej. ✉email: jan.panefarre@chemie.uni-marburg.de; christine.ziegler@biologie.uni-regensburg.de

The stressosome is a megadalton signaling complex first described in the Gram-positive model organism *Bacillus subtilis*, for which a wealth of mutagenesis and functional data are available[1–3]. It consists of multiple copies of a small single STAS (sulfate transporter anti-sigma factor antagonist) domain protein, RsbS, and another STAS-domain protein comprising an additional globular N-terminal sensory domain, RsbR[4]. *Bacillus* species encode several RsbR paralogs, differing in their sensory domains, which are also incorporated into the stressosome[5–9]. However, the nature of the signaling molecule sensed by the RsbR paralogues, with the exception of the blue light responsive YtvA protein, and the nature of RsbR activation remains an open question[8,10]. RsbT, the third protein of the stressosome complex, was suggested to bind to RsbS in the stressosome core in an inactive state[11]. Stress signal perception results in the RsbT-dependent phosphorylation of conserved serine and threonine residues in the STAS domains of RsbS and RsbR paralogs, respectively, and finally the detachment of RsbT[12–14]. Free RsbT initiates a signaling cascade resulting in the up-regulation of stress gene transcription; which, in *Bacillus* and close relatives, is achieved by the activation of the alternative sigma factor, σ$^B$, involving the proteins RsbU, RsbV, and RsbW[14–18] (Supplementary Fig. 1). The activation state of the stressosome is reset by the removal of phosphates from RsbS and RsbR by the phosphatase RsbX, allowing for a new round of phosphorylation[12]. Transcription of RsbX is positively controlled by σ$^B$ and this provides a feedback loop to reset the stressosome to an inactive state[19]. However, precise knowledge on the feedback mechanism to attenuate RsbT release during continuous stress exposure is still scarce.

Three published single particle cryo-electron microscopy (cryo-EM) reconstructions reveal that the *Bacillus* and *Listeria* type stressosomes are assembled in a pseudo-icosahedral scaffold, formed by interaction between the RsbR STAS domain and RsbS[11,20,21]. The STAS domains of RsbS and RsbR dimers constitute the stressosome core with a proposed 20 RsbR$_2$:10 RsbS$_2$ stoichiometry. This stoichiometry was inferred from the mushroom-like structure of the RsbR sensory domains, termed turrets, which protrude out of the STAS domain core[11]. From this stoichiometry, D2-symmetry was superimposed to the icosahedral STAS domain for both *B. subtilis* stressosomes. Interestingly, a recent cryo-EM structure of the *L. monocytogenes* stressosome complex from the icosahedral core resolved to 3.38 Å and a 4.48 Å map without any symmetry imposed (C1 map) confirms the hetero-assembly of one RsbS dimer with two RsbR dimers along the threefold symmetry axis similar to the *B. subtilis* stressosome, but differs with respect to the RsbR:RsbS stoichiometry of the pentameric faces of the STAS domain core, resulting in a breakdown of the D2 symmetry[21]. These differences in stressosome assembly are intriguing, and more structures of stressosome complexes from different organisms are required to understand the role of STAS domain core interactions with RsbT. As STAS domains accommodate the stress-regulated phosphorylation sites, molecular insights into the dynamics of the core are of paramount importance to understand the mechanism of stressosome signal transduction.

Here, we determined a 3.87 Å reconstruction from a D2 symmetrized cryo-EM map of the stressosome of *Listeria innocua* (*Li*RsbRS) assembled from co-expressed *Li*RsbR and *Li*RsbS proteins. While the *Li*RsbR complex confirms the 20 *Li*RsbR$_2$:10 *Li*RsbS$_2$ stoichiometry found in *B. subtilis*, it shows pronounced differences in STAS domain secondary structure and assembly compared to the recently published *L. monocytogenes* structure. Our structure provides insights into the movement of sensor linker helices and STAS domains. We hypothesize a key role for RsbT–RsbT interactions in controlling the allosteric

phosphorylation of RsbS and RsbR proteins thereby exploiting the pseudo-icosahedral symmetry of the STAS domain core. Finally, we propose a mechanism of a stress-induced conformational change in the N-terminal domain of *Li*RsbR, resulting in an altered accessibility of phosphorylation sites.

## Results

**Architecture of the LiRsbRS complex**. Two species of stressosome complex assemblies were eluted at different salt concentrations during ion exchange chromatography: a dimeric and a monomeric form, respectively (Supplementary Fig. 2a). The oligomer complex distribution remained stable at the different salt concentrations during size-exclusion chromatography on a Superose 6 Increase column (Supplementary Fig. 3a), however, complexes self-assembled stronger at higher ion strength judged from negative stain EM (Supplementary Fig. 3b). Only the monomeric complex was subjected to structure determination by single particle cryo-EM after running a size-exclusion chromatography (Supplementary Fig. 2b). Interestingly, the MS analysis on the phosphorylation state of the SEC fraction used for structure determination revealed a phosphorylation of Thr241, while conserved sites Thr175 and Thr209 were not phosphorylated (Supplementary Fig. 2c). Although the stressosome particles were easy to identify in the cryo-EM micrographs, a reconstruction into a 3D volume was challenging due to the pseudo-icosahedral symmetry. 3D classes were selected after classification in RELION not only based on resolution but on the simultaneous appearance of turret densities at a comparably high signal level to the STAS domain core. Classifications that yielded only one or two turrets were rejected independently of the nominal resolution given by the respective programs. Signal subtraction routines for turrets and core were performed for further improvement of the alignment. 3D refinement was performed with the non-uniform refinement method in Cryosparc V2[22], and yielded a map without imposing any symmetry (C1 map to 4.2 Å, Supplementary movie 1), a map with imposed D2 symmetry (D2 map to 3.87 Å, Fig. 1d–f and Supplementary Fig. 5), and a map with icosahedral symmetry (I map to 3.45 Å, Supplementary Fig. 6). The resolution of the side chains in the C1 map was of sufficient quality to unambiguously identify and model the STAS domains of *Li*RsbR and *Li*RsbS (Figs. 1 and 2). The stoichiometry of RsbR:RsbS was identical to the one described for the *Bacillus* stressosome, 20 *Li*RsbR$_2$:10 *Li*RsbS$_2$. In addition, the D2 symmetry axes were visible in the C1 map, which helped improve the turret densities significantly, consistent with the *Bacillus* stressosome assembly. The 10 *Li*RsbS dimers form two hook-shaped strips along the core (Fig. 1a, f in yellow and orange color). This assembly leads to pentameric planes with either 1 or 2 RsbS monomers connecting to 4 or 3 RsbR monomers in one of the 12 edges (Fig. 1b, f). A pentameric plane thereby comprises five triangular faces, all of them consisting of one RsbS and two RsbR dimers (Fig. 1c, f). The stressosome icosahedron is comprised of 20 hetero-triangular faces, which most likely is the functional unit for RsbT interaction. This feature is consistently conserved in the recently published cryo-EM C1 map of the stressosome complex from *Listeria monocytogenes* (EMD-4508; *Lm*RsbRS) determined to a resolution of 4.21 Å (Supplementary Fig. 7). However, in contrast to our *Li*RsbRS and both *Bs*RsbRS structures, the D2 symmetry is broken in *Lm*RsbRS; as *Lm*RsbS dimers are circularly arranged around two pentagons (Supplementary Fig. 7, yellow and orange RsbS circles). In our unsymmetrized *Li*RsbRS map, nearly 20 turrets are present simultaneously in a *B*-factor sharpened map, which highlights the quality of particle alignment. Pronounced differences in the STAS domain dimer interfaces of *Li*RsbR and *Li*RsbS can be clearly identified (Fig. 2a, b) and strengthen the

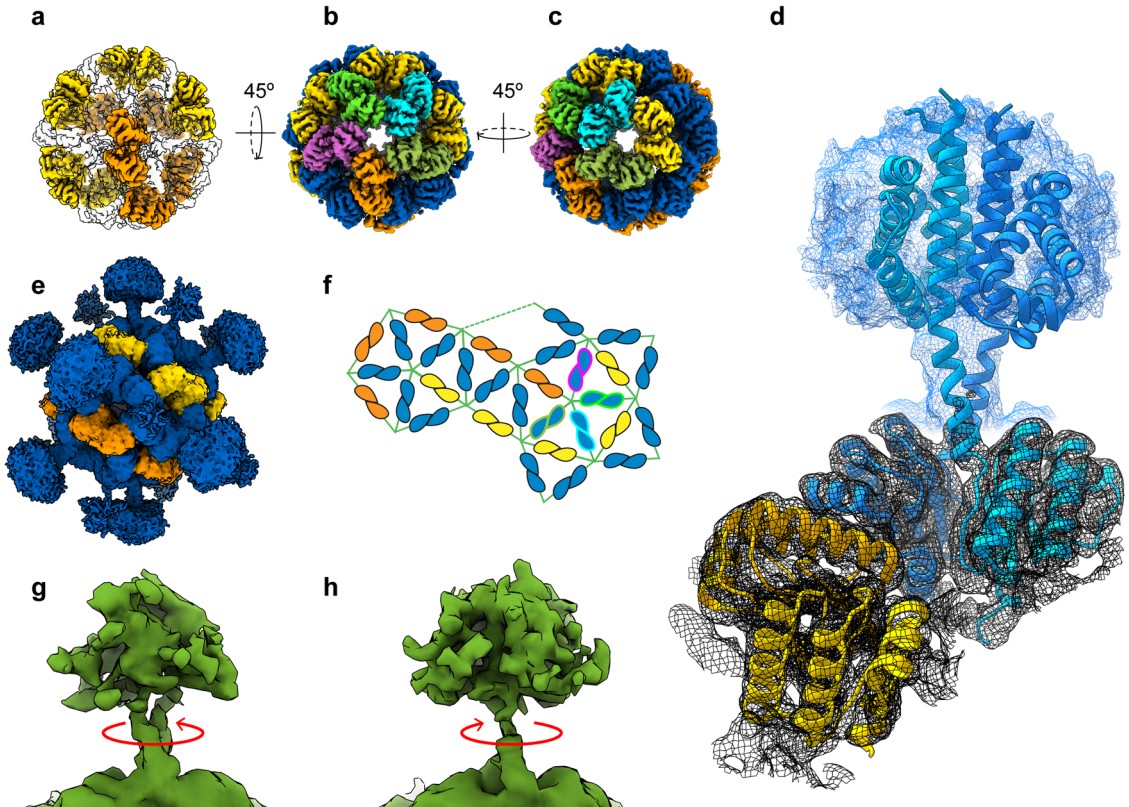

**Fig. 1 Cryo-EM derived reconstructions and models of the *Listeria innocua* stressosome complex (*Li*RsbRS) with imposed symmetry I2 to 3.45 Å (a–c) and in D2 to 3.87 Å (d, f) and fitted atomic models in the D2 map. a** Electronic potential map of the *Li*RsbRS stressosome with I2 symmetry imposed, with *Li*RsbS dimers shown in yellow and orange color. The STAS-domain dimers of the *Li*RsbR are indicated as transparent shapes. A hook-shaped strip of 5 *Li*RsbS dimers (yellow) is consecutively mirrored along perpendicular D2 axes yielding the second strip of 5 *Li*RsbS dimers (orange). **b, c** The STAS-domain dimers represent a leg in the 20 congruent triangular faces. 12 corners connect always five (**b**) of the triangular faces (**c**). Protomers from different dimers shown in different colors. **d** Fitted atomic models of *Li*RsbR and *Li*RsbS in the D2 symmetrized 3.87 Å map, with the *Li*RsbR dimer and *Li*RsbS dimer shown in blue and yellow color, respectively. The map volume is rendered at two levels, focusing at the STAS-core (black) and showing the sensory domain of *Li*RsbR (blue, level was set about a factor 5 higher). **e** In the assembly of the *Li*RsbRS, the *Li*RsbS (yellow and orange color) and the *Li*RsbR subunits (blue color) are each in contact to another. RsbRs in one pentagon are highlighted in similar colors as in (**b, c**). **f** A two-dimensional representation of the pentagonal and triangular faces in the STAS icosahedron colored according to (**b, c**). **g, h** 3D variability analysis in cryoSPARC v2.9[23] in C1 created a set of 40 continuous 3D volumes with two distinct conformations of the RsbR sensor domain are visible along the linker helix interface of the RsbR dimer: slightly coiled conformation (**g**), parallel straight conformation (**h**), with the former predominantly selected in 3D classification of the D2 map (**d**).

assignment of R and S components within the *Li*RsbRS complex (Fig. 1e, f). For the published *Lm*RsbRS complex, the inconsistent signal intensity of well-resolved STAS domain core vs. weakly resolved *Lm*RsbR turrets (Supplementary Fig. 7) might have hampered the unambiguous assignment of *Lm*RsbR and *Lm*RsbS. Furthermore, the *Lm*RsbRS complex was assembled after purification, which could affect complex assembly, while both *Bacillus* and *Li*RsbRS stressosomes were co-expressed and co-purified.

3D variability analysis in cryoSPARC v2.9[23] in C1 was used to investigate the conformational heterogeneity in the stressosome cryo-EM data set. From a group of 40 continuous 3D volumes (Supplementary movie 2), two distinct conformations of the RsbR sensor domain are visible (Fig. 1g, h), while the STAS domain adopts one stable conformation within the stressosome core. Conformational changes in the sensor domain occur along the linker helix dimer interface of RsbR. The linker helices switch from a straight orientation to a wedged, slightly intertwined conformation (Fig. 1d). The conformational changes in the linker helices are accompanied by changes in the sensor domain; however, due to the reduced resolution in this domain, it was not possible to model these changes with confidence. However, as the STAS domains show one distinct conformation we conclude that

the sensor domain switches conformations stochastically when RsbT is not bound to the stressosome. These conformational changes of the RsbT-unbound N-terminal domains do not seem to affect the conformation of the STAS domain, which suggests an additional element is required for communicating sensor-domain changes to the phosphorylation status of the stressosome STAS domain (Fig. 3a, b).

**Models of the LiRsbR and LiRsbS STAS domains.** Modeling of *Li*RsbR (Fig. 2a) and *Li*RsbS (Fig. 2b) confirms the high conservation of the STAS domain fold (Fig. 2c). *Li*RsbR enters the STAS domain as the linker helix ($\alpha_0$), while a beta-strand ($\beta_0$) is found at the N-terminal end in *Li*RsbS. $\alpha_3$ is well-resolved in *Li*RsbRS (Supplementary Fig. 8d, e), while it is not resolved in *Lm*RsbR. In addition, the models of the STAS domains of *Li*RsbR and *Lm*RsbR (pdb entry 6QCM) differ by a frame shift of 6 residues (Supplementary Fig. 8a) due to a different positioning of Pro1β56 and Pro159 (Supplementary Fig. 8b, c). The conserved *Li*RsbS Ser56 and *Li*RsbR Thr175 and Thr209 are not phosphorylated in our structure. Close to Thr241 in *Li*RsbR, a spherical density was assigned as phosphoryl group in accordance with the ESI–MS–MS (Fig. 2d; Supplementary Fig. 2c).

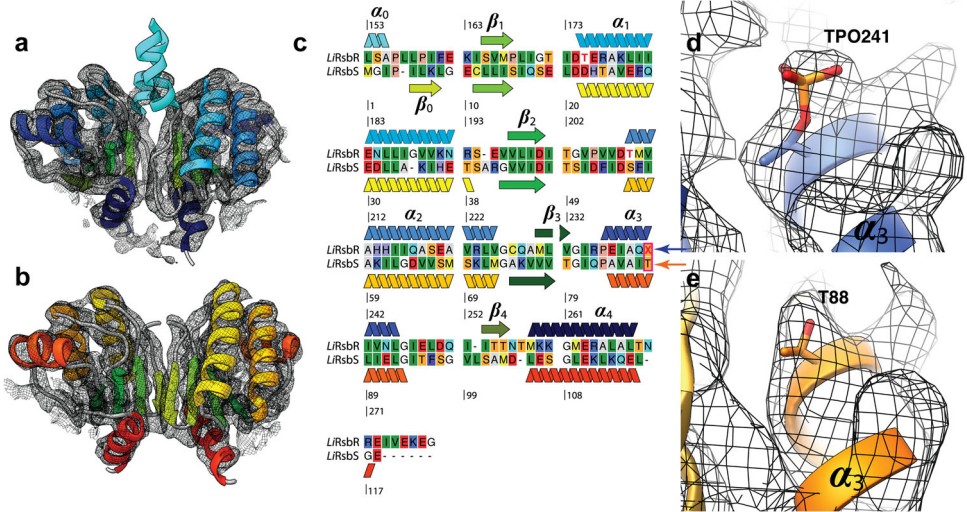

**Fig. 2 Model of the STAS-domains in the stressosome complex from *L. innocua*. a** *Li*RsbR and **b** *Li*RsbS. The helices of *Li*RsbR, $\alpha_0$–$\alpha_4$ are shown in different shades of blue and the $\beta$-sheets, $\beta_0$–$\beta_4$ in shades of green, respectively. In *Li*RsbS the helices are color-coded in yellow/orange and $\beta$-sheets in shades of green. The D2 cryo-EM map is shown as black mesh. The density of $\alpha_0$ was very pronounced and allowed for an unambiguous assignment of the STAS core with the stoichiometry of 20 *Li*RsbR$_2$: 10 *Li*RsbS$_2$ described in Fig. 1. **c** The topology of (**a**, **b**) is shown in the sequence alignment of *Li*RsbR and *Li*RsbS and Supplementary movie 3. X indicates the phosphorylated threonine residues. The density map and model of the region of $\alpha_3$ containing the conserved threonine adjacent to the phosphorylation site (**d**) in a phosphorylated state in RsbR (**e**) in an unphosphorylated state in RsbS.

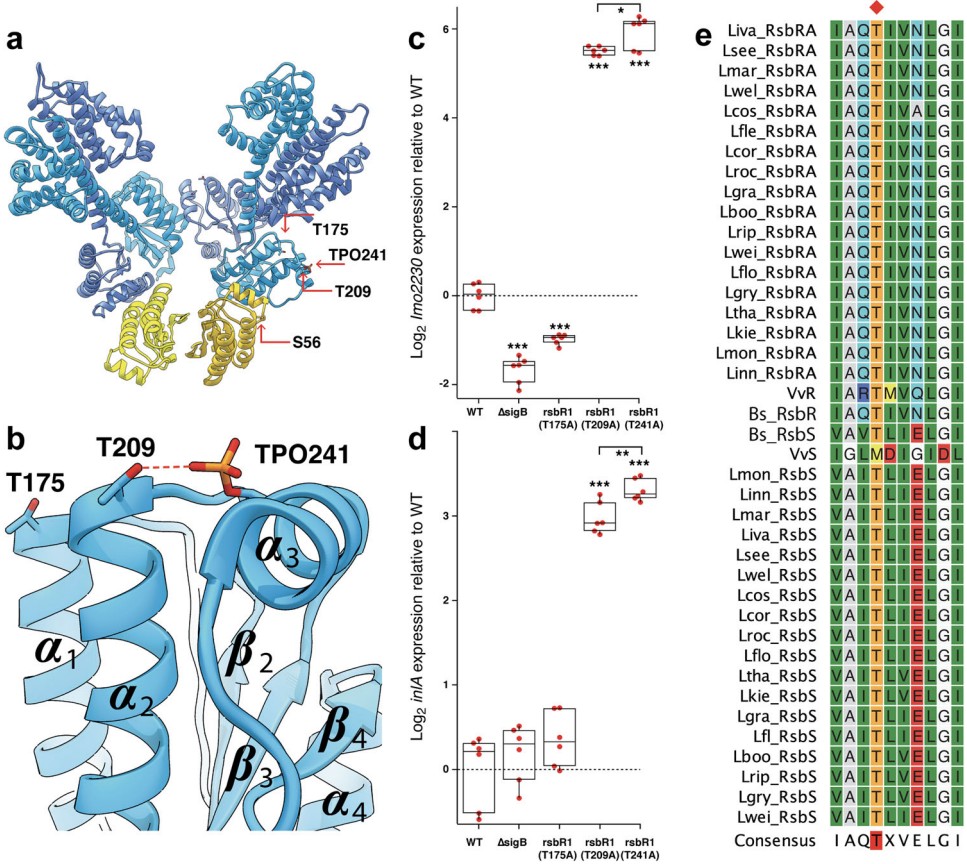

**Fig. 3 Phosphorylation sites and their surroundings in *Li*RsbRS. a** The heterotriangle formed from two dimers of RsbR and one dimer of RsbS, with the regulatory phosphorylation sites and their interaction partners indicated by red arrows. **b** a close-up of the RsbS interaction surface, with Ser56 shown in $\alpha_2$ and the Thr88 in $\alpha_3$. **c** Relative expression of the SigB-dependent genes *lmo2230* (**c**) and *inlA* (**d**) in unstressed *L. monocytogenes* cells relative to wild type in mutants of *rsbR1* and *sigB*. The mean value is shown by the horizontal line in the boxes, and the height of the box indicates standard deviation, n = 6 for each experiment. **e** Alignment of STAS domain proteins found in different Gram-positive bacteria, with the conserved T241 residue indicated with a red diamond. The raw data for panels c and d can be found in Supplementary Data 1.

Such a density was not observed in *Li*RsbS at Thr88 (Fig. 2e). Thr241 in *Li*RsbR and the corresponding Thr88 in *Li*RsbS are conserved in the STAS domains of RsbR and RsbS of different *Listeria* species as well as in *Bacillus subtilis*.

The phosphoryl-group brings Thr241 in interaction range to Thr209 (Fig. 3a, b). At this point we can only speculate that TPO241–Thr209 is mimicking a potential interaction between both residues, which would have taken place with the vice versa phosphorylated Thr209 (TPO209–Thr241). A similar bridged interaction, Thr209–PO$_3$–Thr241, would be conserved by Thr205–PO$_3$–Thr237 in *B. subtilis*. Substitution of either T209 or T241 with alanine in *L. monocytogenes* resulted in the same phenotype, with increased transcription of two σ$^B$ regulated genes, *lmo2230* and *inlA*[24,25], during exponential growth in a complex broth medium (Fig. 3c, d). This result suggested that loss of this T209–PO–T241 interaction resulted in an inability to suppress σ$^B$ activity in the absence of environmental stress. In addition, exploring the sequences of STAS domain proteins of related bacterial species indicates that T241 is a conserved residue, at least in Gram-positive bacteria (Fig. 3e).

**Interdimeric interactions within the STAS core**. We observe different phosphorylation states of Thr241 depending on their interaction with their STAS domain neighbors within the pseudo-icosahedral assembly (Fig. 4 and Supplementary Fig. 10). Phosphorylated Thr241 was only observed when RsbR was interacting with RsbS (Fig. 4c, d) not in an RsbR–RsbR interaction (Fig. 4g, h), which was instrumental in resolving the phosphoryl density during processing. The interaction between the two *Li*RsbR dimers (Fig. 4g, h) is comparable, but not similar to the interaction of *Li*RsbS with *Li*RsbR at the R–S and S–R interface, respectively (Fig. 4c–f). While the interactions between R–R and R–S comprise similar structural elements, the interactions at the S–R interface are reduced just to a hydrophobic coordination of Leu224 by Leu100 and Ile90 in *Li*RsbS (Fig. 4h). Therefore, the R–S and S–R interface differ with respect to the coordination and flexibility of $a_3$ and loop $a_3$–$β_4$. Direct consequence of Thr209–TPO241 is a conformational change in $a_3$ and subsequently in symmetry break in the hetero-triangular face (Fig. 4).

**RsbT binding and release in LiRsbRS**. We have purified *Li*RsbT after expression in *E. coli* and performed an in vitro phosphorylation of *Li*RsbRS complex (Supplementary Fig. 11a, b). We observe a bi-phasic phosphorylation time-course. *Li*RsbS phosphorylation increases linearly in the first time points during 1 min, while phosphorylation of *Li*RsbR starts only after this linear phase. Interestingly, when *Li*RsbR increases *Li*RsbS phosphorylation jumps into saturation suggesting that RsbT binding to *Li*RsbR and *Li*RsbS is not independent to each other. We performed an additional size-exclusion chromatography run with the *Li*RsbRS complex incubated with RsbT (Supplementary Fig. 11c). Although we obtained a ternary complex the amount of RsbT was varying between different experiments reflecting a highly dynamic interaction in vitro like what was observed for *Lm*RsbRST binding[21]. A negative stain analysis of the co-eluted ternary complex suggested additional density for RsbT (Supplementary Fig. 11d), however, we were not able to obtain a stable *Li*RsbRST complex suitable for high-resolution structure determination.

To investigate the interaction of RsbT with the RsbRS complex we performed a docking study using two structurally well described STAS domain complexes, the crystal structures of SpoIIAB–SpoIIAA (pdb entry code 1TIL and 1TH8) and of RsbV–RsbW (pdb entry code 6M37)[26], as blueprint. A homology model of ATP and ADP bound *Li*RsbT was generated and docked

to *Li*RsbR (Fig. 5a, b) and *Li*RsbS (Fig. 5c) using the spatial orientation of SpoIIAA to SpoIIAB and RsbV and RsbW, respectively, as a template: the STAS domain of *Li*RsbR/S aligned with SpoIIAA/RsbV and ATP/ADP-bound *Li*RsbT aligned with SpoIIAB/RsbW. The LiRsbT models were docked to LiRsbR (Fig. 5a, b) and LiRsbS (Fig. 5c, d) using the spatial orientation of SpoIIAA to SpoIIAB as a template: the STAS domain of LiRsbR/S aligned with SpoIIAA and ATP/ADP-bound LiRsbT aligned with SpoIIAB. The superposition of SpoIIAA and LiRsbR (Fig. 5b) revealed that due to the interaction of T209–TPO241, $a_3$ has moved towards $a_2$ in an 'up-conformation' compared to the $a_3$-'down-conformation' in a non-phosphorylated RsbS (Fig. 5c, d). The 'up-conformation' will result in a steric clash of ADP-bound LiRsbT and $a_3$ (Fig. 5b; clash indicated by red star). By contrast, docking of ATP-bound LiRsbT to non-phosphorylated LiRsbS (Fig. 5c) shows that Ser56 is accessible for phosphorylation by LiRsbT, as no interaction between Ser56 and Thr88, the structural equivalent of Thr241 in LiRsbR (Fig. 5d), takes place and $a_3$ thus remains in a 'down-conformation'. In summary, $a_3$ and loop $a_3$–$β_4$ must move during RsbT release. We suggest that phosphorylation of RsbR affects the conformational flexibility of $a_3$ via Thr241.

**Docking of multiple LiRsbT to the LiRsbRS complex**. Multiple copies of *Li*RsbT were docked to the entire stressosome complex (Fig. 6a) to investigate steric clashes. In the functional hetero-triangular face, one *Li*RsbT protein can bind to every STAS domain monomer without steric clashes (Fig. 6b). *Li*RsbR would have sufficient space to capture *Li*RsbT. *Li*RsbR shows a mainly negatively charged surface, while one side of *Li*RsbT is mainly positively charged (Fig. 6d; coulombic surface Supplementary Fig. 9a). The observed conformational change of the linker helices observed by the 3D variability study (Fig. 1g) would most likely affect the binding pose and binding affinity of RsbT and RsbR. A comparison of the electrostatic surfaces obtained from homology models of the other three RsbR paralogs in *L. innocua*, which are found outside the σ$^B$-operon, reveals a significant variation (Supplementary Fig. 12a–d) despite their high degree of conservation in the STAS domains (Supplementary Fig. 12e). The sensory turrets and STAS domain in *Li*RsbR form a clamp-like binding pocket in which *Li*RsbT would fit in a fixed orientation, but with the ATP-binding site not accessible to Thr209. Having *Li*RsbT docked at this position to *Li*RsbR would coordinate *Li*RsbT binding to *Li*RsbS (Fig. 6d). For the proposed *Li*RsbST complex, the target Ser56 and the potential catalytic base Glu45 align for phosphorylation (Fig. 6e). In contrast to SpoIIAB, a positively charged residue (SpoIIAB-Arg105) for electrostatic stabilization of the transition state is not conserved in *Li*RsbT. In the proposed *Li*RsbRS–RsbT assembly the ATP-lid is stabilized by the interaction with a neighboring *Li*RsbT (bound to *Li*RsbS). The order of binding to *Li*RsbR might be restricted such that *Li*RsbT can only bind to *Li*RsbR if *Li*RsbT is either not yet bound to *Li*RsbS or has been released. The observed conformational changes in the sensor domain and linker helices might help re-orient *Li*RsbT on *Li*RsbS for productive phosphorylation.

**Discussion**

The stressosome transduces environmental stress signals to one single cellular output, which is the phosphorylation-dependent release of the kinase RsbT to start the downstream sigmaB-signaling cascade[14,17]. This represents a straightforward 'actio-reactio' principle: increased amount of stress results in a higher number of released RsbT proteins. For *B. subtilis*, it was suggested that RsbS preferably sequesters RsbT in the *Bs*RsbRS complex[23].

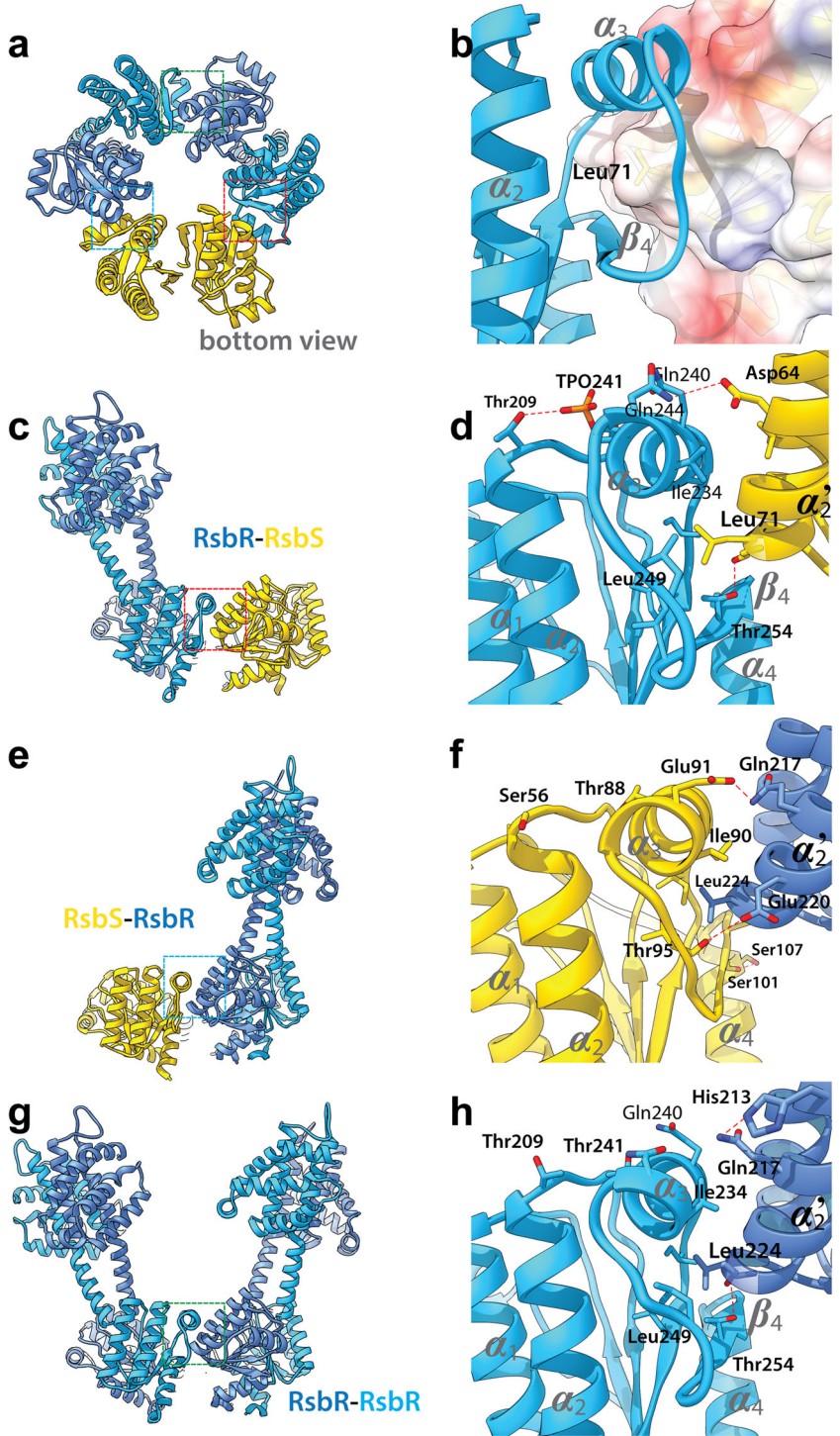

**Fig. 4 Interdimer interaction surfaces in the RsbR:RsbS heterotriangle.** The three unique interaction surfaces can be seen from the bottom in (**a**) and from the side in (**c**), (**e**) and (**g**). A surface rendering of RsbS and the sidechain in the hydrophobic pocket can be seen in (**b**). Leu71 from the $\alpha_2$ helix of RsbS can insert into the hydrophobic pocket formed by $\alpha_3$ and $\beta_4$ of RsbR (**d**), the interaction further strengthened by hydrogen bonds formed between Glu91 of RsbS and Gln217 on RsbR. A very similar interaction occurs between $\alpha_2$ of RsbR and the adjacent RsbS (**f**) or RsbR (**h**) protomer, where Leu224 inserts into the hydrophobic pocket former by $\alpha_3$ and $\beta_4$ of the adjacent STAS domain protein.

In the *Bs*RsbRST structure, 20 *Bs*RsbT molecules were thus positioned to the 20 *Bs*RsbS molecules[11]. However, environmental stresses lead to an increase in RsbT kinase activity against RsbR, too[13], which is not accounted for in an only RsbS–RsbT interaction. The *Li*RsbRS complex now presented in this manuscript provides a unique opportunity to observe conformational

changes in response to differences in phosphorylation especially in RsbR. Different to the recently published structures of the *Listeria monocytogenes* complex[21], we were able to resolve helix 3 and its significant conformational changes. In contrast to the recent *B. subtilis* structure[20], we present a more dynamic picture to the STAS–STAS domain interface and its role in signal

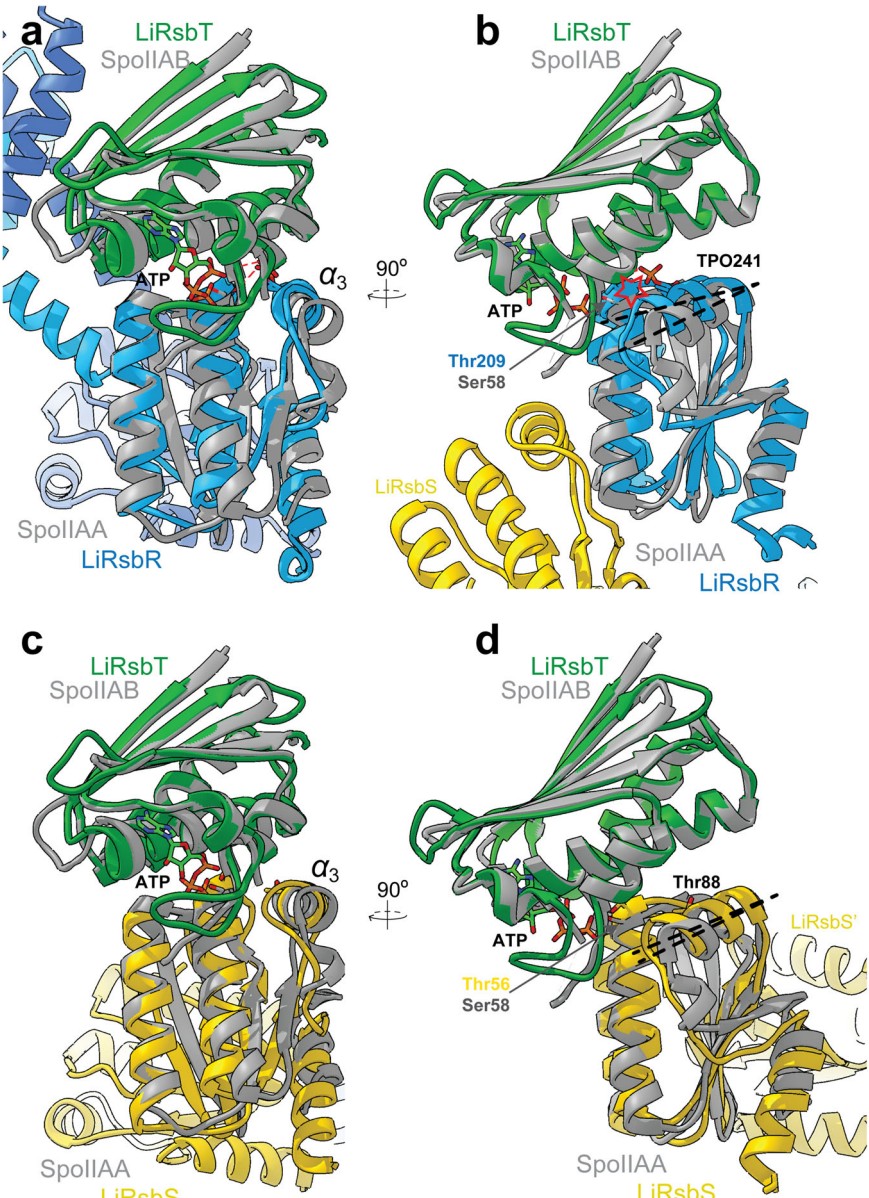

**Fig. 5 Docking of *Li*RsbT to *Li*RsbS and *Li*RsbR. a** Superposition of *Li*RsbR model (in blue color) with a crystal structure of SpoIIAA (gray) in complex with the ADP-bound SpoIIAB (pdb entry 1TH8 gray color). A homology model of ADP-bound *Li*RsbT (in green color) is aligned matching the orientation in the SpoIIAB complex, while SpoIIAA is aligned to the STAS domain of *Li*RsbR. **b** Close-up view of the STAS domain superposition in (**a**) rotated by 90° (color coding as in **a**). Phosphorylation sites and the respective interacting sidechains are shown as stick models (carbon atoms in blue or yellow, respectively, oxygen atoms in red, and nitrogen atoms in dark blue). The relative conformational difference between the not phosphorylated SpoIIAA from *B. subtilis* and the phosphorylated *Li*RsbR is indicated by the dashed lines, and the model clash indicated by a red star. **c** Superposition of *Li*RsbS model (in yellow color) with the SpoIIAA in complex with ATP-bound SpoIIAB (in gray color). A homology model of ATP-bound *Li*RsbT (in green color) is aligned to *Li*RsbS matching the orientation of SpoIIAB in the SpoIIAA:SpoIIAB in complex. A 90˚ rotated view can be seen in (**d**).

propagation in the complex. The high quality of the D2 map with a completely resolved STAS domain and linker helix allow for insights on how the stress signal is translated into a conformational change upon activation. We suggest that RsbS still acts as a primary binding site, lacking the steric hindrance of the RsbR turrets, which is supported by the in vitro phosphorylation of the *Li*RsbRS complex (Supplementary Fig. 11). It requires a certain threshold of phosphorylated *Li*RsbS before phosphorylation of *Li*RsbR is detected. We assume that at least one RsbT is bound to the RsbR protomer in direct contact with RsbS. Without stress RsbT is not able to phosphorylate RsbR. The electrostatic surface potential representation of *Li*RsbR suggests an ionic interaction of

the negatively charged *Li*RsbR dimer with the positively charged *Li*RsbT, coordinated by the RsbR STAS domain of one protomer and the turret of the other protomer within the RsbR dimer in a clamp-like fashion. The candidate *Li*RsbR paralogs Lin0204, Lin1683, and Lin1956, display less pronounced negative charge patches in the same region (Supplementary Fig. 12). Differences in electrostatic attraction between RsbR and RsbT and variation of the affinity and/or number of RsbT molecules might cause the modulation of the stress downstream signal when the stressosome is assembled from different RsbR paralogs.

RsbT might also have an impact on stabilization and signal transduction within the whole stressosome complex. The high

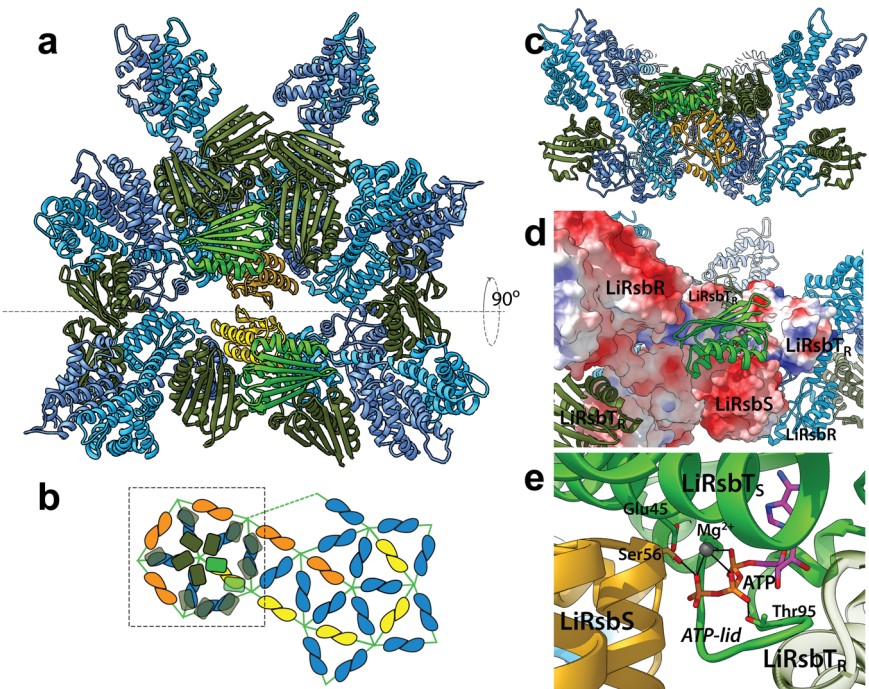

**Fig. 6 Proposed assembly of the functional hetero-triangular *Li*RsbRS complex with docked *Li*RsbT. a** View on the triangular faces of *Li*RsbRS (RsbR in blue and RsbS in yellow colors), and with *Li*RsbT homology models (shown in dark green color docked to *Li*RsbR and in light green color docked to *Li*RsbS). **b** Two-dimensional representation of the pentagonal and triangular faces in the STAS icosahedron colored according to (**a**). The dashed box indicates roughly the composition shown in (**a**). **c** Sideview (as indicated by the gray, dashed line in **a**) on the *Li*RsbRS–RsbT assembly. In this assembly, only the *Li*RsbT bound to an *Li*RsbS can dissociate from the complex; *Li*RsbTs bound to *Li*RsbRs are sterically confined by the linker helices. **d** Vicinity of the *Li*RsbT$_S$ (bound to *Li*RsbS, light green) binding pose shown as surface colored according to the electrostatic potential. The *Li*RsbT-binding site is created by equal contributions of one *Li*RsbS and two *Li*RsbTs, some contact to the sensory domain of *Li*RsbR is also feasible. **e** Vicinity of ATP in the proposed *Li*RsbS–*Li*RsbT complex. The target Ser56 and the potential catalytic base Glu45 are indicated. In the proposed *Li*RsbRS–RsbT assembly the ATP-lid is stabilized by the interaction with a neighboring *Li*RsbT$_R$ (bound to *Li*RsbR).

resolution of the *Listeria innocua* stressosome STAS domain shows that in the up conformations of $\boldsymbol{a}_3$ (in RsbR) RsbT cannot be docked in way that its ATP-binding site would be accessible to the phosphorylation sites. Vice versa, in the down conformations of $\boldsymbol{a}_3$ in the de-phosphorylated state (in RsbS) docking is possible. The $\boldsymbol{a}_3$–$\boldsymbol{\beta}_4$ delineated pocket is the essential interface for inter-action between the STAS domains of adjacent dimers, and the nature of the interactions between the domains is very similar to that found in *B. subtilis*, i.e. the major interaction is between hydrophobic residues, with additional, less conserved residues mediating electrostatic interactions between the protomers between the $\boldsymbol{a}_3$–$\boldsymbol{\beta}_4$ of one domain and α2 of the other[20]. The two distinct conformations of the sensor domain and linker helices (straight and intertwined in Fig. 1g) could be attributed to dif-ferent phosphorylation states of RsbR. However, the STAS domains do not change conformation in the 3D variability study (Supplementary movie 2), which would suggest a rather rigid stressosome core of *L. innocua*. It is more likely that without RsbT bound, the N-terminal sensor domains display an intrinsic flexibility.

Thr205 in *Bs*RsbR (corresponding to Thr209 in *Li*RsbR) is only phosphorylated after severe stress in order to limit the stress transmission process[27]. Therefore, we assume that RsbT can bind to RsbR in different conformations without having the ATP-site positioned for phosphorylation. Our docking studies on *Li*RsbS suggest that an activation of *Li*RsbS Ser56 requires neighbor RsbTs to stabilize $\boldsymbol{a}_3$ and loop $\boldsymbol{a}_3$–$\boldsymbol{\beta}_4$ in RsbS and the ATP-lid in RsbT. RsbT is missing the positive charges to stabilize the binding pocket for ATP by its own. A computational study on the *Bs*RsbRST complex supports this notion and concluded that a

model in which phosphorylation by RsbT is increased by the presence of phosphorylated neighbors provided the best fit to the experimental data[28]. We propose that a direct interaction of RsbT with RsbR already in the down-regulated state will increase the RsbT kinase activity on RsbS and represents the first step in the allosteric phosphorylation process. As a consequence, neighbor-ing RsbR dimers decorated with RsbT would provide additional coordination for the RsbT–RsbS binding increasing the affinity of RsbS towards RsbT.

Different RsbR paralogs with different RsbT affinities, because of their electrostatic potential, would provide an elegant way to modulate efficiency of the RsbT kinase reaction. Most interesting, recent data on *B. subtilis* RsbR paralogs revealed a similar spec-trum of stress stimuli sensed by the paralogs RsbRA-D, however with varying response profiles[29].

By including RsbR–RsbT and RsbT–RsbT interactions we describe the activation mechanism as following: A stress stimulus perceived by RsbR turrets triggers the re-orientation of RsbT necessary to phosphorylate the target Ser/Thr, because con-formational changes will be transmitted via the linker helices by a RsbR–RsbT interaction affecting subsequently the RsbT–RsbS interaction. Conformational changes in the RsbR turrets and the linker helices might result directly in an altered RsbR–RsbT interaction, which in turn affect the RsbT–RsbS interaction (T–T crosstalk). Alternatively, conformational changes in the STAS domain of RsbR, can be transduced via the RsbR–RsbS inter-dimeric interactions in order to render Ser/Thr residues accessible for RsbT to phosphorylate (R–S crosstalk). RsbT will subse-quently be released from RsbS by a conformational change in the $\boldsymbol{a}_3$ helix and loop $\boldsymbol{a}_3$–$\boldsymbol{\beta}_4$, as we have observed in the RsbR dimers

exhibiting the Thr209-TPO241 interaction. The flexible nature of $Lm$RsbR $a_3$ is in line with the potential regulatory role of this region[21]. Once RsbT is released from RsbS, a re-arrangement of RsbT bound to RsbR may occur. If stress continues or increases, RsbT is already positioned to phosphorylate RsbR T209 and will be released. In the allosteric context, release of RsbT from RsbR would lead to a decrease of activation of the RsbT positioned to phosphorylate RsbS and an attenuation of the transmission process. The stepwise re-orientation of RsbT would agree with the observation that binding of RsbT to RsbR increases the phosphorylation probability of RsbS[30] but that phosphorylation of RsbR at Thr209 decreases stress signal transmission[27]. Our model can be extended to the recovery following a stress activation event. We assume that RsbT would dissociate from RsbS first, allowing the access of RsbX to dephosphorylate RsbS. A still bound RsbT to the adjacent RsbR would not hinder such a RsbS–RsbX interaction and dephosphorylation of RsbS may occur before RsbT is released from RsbR. The dephosphorylation of RsbS releases the tension on α2-helix of the neighboring RsbR connected to RsbS (Fig. 4f). Dephosphorylation of S56 in RsbS would support the conformational transition of α3-helix in RsbR from the 'up' to the 'down-state'. In further events, RsbX would dephosphorylate RsbR, resetting the stressosome to process imminent stress.

Different mutations in the RsbR N-terminal domain, e.g., $Bs$RsbR E136K significantly increases the basal level of the $\sigma^B$ response, even without stress[10]. However, an RsbS mutation with S59A negates these effects suggesting that N-terminal mutations in RsbR lead to increased phosphorylation in RsbS[10]. According to our docking model, Lys141 of $Li$RsbR, positioned in the RsbR linker helix, would contact Gln20 in RsbT and, in a putative $Li$RsbR E140K mutation corresponding to $Bs$RsbR E136K, Lys141 would point in a different direction disrupting the RsbT–RsbR interaction. It can be noted that previously described mutations in the $Bs$RsbR N-terminal domain would be positioned in the RsbR–RsbT contact area[4]. The postulated T–T cross talk would also shed light onto the puzzling symmetry breaks in RsbR–RsbS assembly. We assume that the key to different stressosome activation profiles is a varying arrangement of different RsbR paralogs within the functional important hetero-triangular faces affecting the RsbT binding/release.

## Methods

**Cloning and heterologous expression of stressosome proteins**. The *L. innocua* genes CAC96120.1 (*Li*RsbR) and CAC96121.1 (*Li*RsbS) were cloned into the pET11a expression vector (Novagen, 69436-3) via NdeI and BamHI restriction sites. A ribosome binding site (construct from GenScript) was inserted between the coding sequences. The construct was transformed into *E. coli* DH5α for plasmid isolation and maintenance.

For over-expression, the pET11a plasmid was transformed into *E. coli* BL21 Star (DE3) cells. Expression cultures were grown to a starting OD$_{600}$ of 0.6–0.8, induced with IPTG (final concentration 1 mM) and harvested by centrifugation after 3 h of expression at 37 °C and 120 rpm. The cells were harvested by centrifugation in a JLA-8.1000 rotor (Beckmann Coulter) at 4000 rpm for 30 min, at 4 °C and stored at −80 °C. For *Li*RsbT a pGEX6P-2 vector (GE Healthcare) containing a GST-tag fused *rsbT* gene from *L. innocua* (CAC96122.1) was transformed into *E. coli* BL21 (DE3) cells and grown in LB media at 37 °C, 120 rpm until OD$_{600}$ = 0.6, then induced with 0.1 mM IPTG and allowed to express for 3 h. The cells were resuspended in Elution buffer (50 mM Tris–HCl pH 7.5, 150 mM NaCl) and lysis and clarification carried out as above.

**Protein purification**. To purify the *Li*RsbRS complex, cells were resuspended in lysis buffer (50 mM Tris–HCl pH 8.3, 10 mM EDTA), broken in a cell disruptor, and cell debris was removed by ultracentrifugation at 150,000 × $g$. Intact stressosome complexes were purified by two subsequent ion exchange steps (50 mM step gradient (50–750 mM NaCl) on a DEAE sepharose column (Cytiva Life Sciences), 60 mL linear gradient (0–750 mM NaCl) on a Resource Q column), followed by size-exclusion chromatography on a Superose 6 Increase column in 100 mM NaCl, 50 mM Tris–HCl pH 8.5.

To purify *Li*RsbT, clarified cell lysate was loaded on a Pierce™ Glutathione Agarose (Thermo Fisher Scientific) gravity column, and eluted with buffer containing 10 mM reduced glutathione. The protein was then pooled and mixed with Prescission Protease, and incubated overnight at 4 °C. Afterwards, the protein was concentrated and purified via SEC chromatography on a Superdex S75 10/300 column.

**Phosphorylation state and salt dependence of the LiRsbRS stressosome assembly**. The purification protocol consisted of anion exchange chromatography using a DEAE resin (Supplementary Fig. 2a) followed by a polishing anion exchange step with Q-resource resin and a final size-exclusion chromatography (SEC) run (Supplementary Fig. 2b). Nearly two-thirds of the *Li*RsbRS complex eluted at 50 mM NaCl in the first DEAE run (Supplementary Fig. 2a). The two fractions from the DEAE run differed in the oligomerization behavior. Complex formation of the first fraction of *Li*RsbRS was confirmed by negative stain EM, revealing the dimerization of two stressosome complexes via two turrets (Supplementary fig. 2a, lower panel). To determine whether ionic strength of the buffers used in anion exchange chromatography affected the stability of the complex, SEC was performed at varying salt concentrations and the individual peak fractions were analyzed by negative stain EM (Supplementary Fig. 3). While the elution volume remained constant regardless of salt concentration, we observed increased interactions of *Li*RsbRS complexes, resulting in string-like assemblies, in response to increasing ionic strength in negative stain EM (Supplementary Fig. 3b).

For comparison of the stability of *Li*RsbRS at different salt concentrations, the complexes were expressed and purified as described above. However, after the first ion exchange step, suitable fractions were concentrated using a 100,000 Da MWCO cut-off centrifugal spin filter (Merck Millipore). The same spin filter was used to exchange the sample buffer by repeated concentration and dilution in the appropriate buffers with varying salt concentrations (100 mM NaCl–1000 mM NaCl), and analytical size-exclusion chromatography was performed on a Superose 6 Increase 10/300column in the respective buffers.

**Cryo-electron microscopy**. For cryo-electron microscopy, 3.5 µL of purified *Li*RsbRS complex at a concentration of 0.6 mg/mL (after SEC) were vitrified on a glow-discharged Quantifoil R2/2 holey carbon grid in a Vitrobot plunge freezer after 3 s of blotting. Movies were collected on a Titan Krios G3 300 keV electron microscope equipped with a Falcon III direct electron detector at a ×75,000 magnification. In total, 1803 movies were collected with 25 frames per movie, at an electron dose 2e−/Å² per frame. For data pre-processing, MotionCor2 and CTFFIND4 were used[31,32] Afterwards, 105,020 particles were picked, extracted and classified in RELION 3.0[33,34]. After 6 rounds of 2D classification, a total of 53,487 particles were selected for further processing. An initial model was reconstructed using RELION 3.0, and all the particles were first subjected to global angular search three-dimensional (3D) classification in 60 iterations with four classes and step size of 7.5°, then the particles from the best class was subjected to the 2nd round of 3D classification with four classes and 3.75° local angular search step in 60 iterations. At this stage, 32,031 good particles were combined from the best 2 classes for the further 3D auto refinement. The 3D refinement was done by the non-uniform refinement in Cryosparc V2[22] using the corresponding map from 3D classification as the reference. Details on the processing workflow are given in Fig. S4 and in Table 1.

**Model building**. For model building of the *Li*RsbR and *Li*RsbS STAS domains, the RsbS anti-sigma-factor antagonist from *Moorella thermoacetica* (pdb 3ZXN) was used as a template, sharing sequence identities of 25% (*Li*RsbS) and 22% (*Li*RsbR), respectively. The D2 map was chosen for model building. The model was then built by manually docking the homology models in Chimera, then adjusted using Coot and refined using Phenix.

**Cultivation and construction of genetically modified L. monocytogenes**. *L. monocytogenes* EGD-e (serovar 1/2a) and *E. coli* DH-5α, plasmids and primers used in this study are listed in Supplementary Tables 1 and 2. Strains were grown in BHI broth or agar (LabM) at 37 °C with constant shaking at 150 rpm min−1. Cells were grown for 16 h and further diluted in fresh BHI to an initial OD$_{600}$ = 0.05 and allowed to grow until mid-log phase (OD$_{600}$ = 0.5). The following antibiotics were added to the media to a final concentration when required: erythromycin (Ery) at 5 µg ml−1 for *L. monocytogenes* strains and ampicillin (Amp) at 100 µg ml−1 for *E. coli* strains.

The gene *rsbR1* were mutated to incorporate Thr-to-Ala in the codons 209 and 241. The codon 209 CAA (Thr) in *rsbR1* was changed to GCT (Ala) and codon 241 ACA (Thr) was changed to GCT (Ala). In both cases silent mutations were added the two adjacent codons in order to discern mutant from WT codons by PCR during mutagenesis while taking in consideration the codon frequencies in *L. monocytogenes* EGD-e strain (*rsbR1* Thr209 -TTGATACAATGGTTG- to *rsbR1* Thr209Ala -GTAGACGCTATGGTA- and *rsbR1* Thr241 -GTTGATACAATGGTT- to *rsbR1* Thr241Ala -GTAGACGCTATGGTA-). The mutagenic sequences, each with a total length of 612 bp, which include SalI and BamHI restriction sequences in each edge, were artificially synthesized in the vectors pEX-K168::*rsbR1* (T209A) and pEX-

**Table 1 Cryo-EM data collection, refinement and validation statistics.**

| | *Li*RsbRS(EMDB-11971) (PDB 7BOU) |
|---|---|
| *Data collection and processing* | |
| Magnification | 75,000 |
| Voltage (kV) | 300 |
| Electron exposure (e⁻/Å²) | 50 |
| Defocus range (μm) | 1.0–1.8 |
| Pixel size (Å) | 1.064 |
| Symmetry imposed | D2 |
| Initial particle images (no.) | 105,020 |
| Final particle images (no.) | 32,031 |
| Map resolution (Å) | 3.8 |
| FSC threshold | 0.143 |
| Map resolution range (Å) | 3.5–7.5 |
| *Refinement* | |
| Model composition | |
| Non-hydrogen atoms | 105,492 |
| Protein residues | 13,320 |
| *B* factors (Å²) | |
| Protein | |
| R.m.s. deviations | |
| Bond lengths (Å) | 0.004 (0) |
| Bond angles (°) | 1.029 (28) |
| Validation | |
| MolProbity score | 2.11 |
| Clashscore | 14.97 |
| Poor rotamers (%) | 0.3 |
| Ramachandran plot | |
| Favored (%) | 91.96 |
| Allowed (%) | 8.01 |
| Disallowed (%) | 0.03 |

A128::*rsbR1* (T241A) (Eurofins Genomics). The mutagenic sequences were subsequently cloned into shuttle vector pMAD, originating pMAD::*rsbR1* (T209A) and pMAD::*rsbR1* (T241A). *L. monocytogenes* electrocompetent cells were created as previously described[35] Briefly, cells were grown at 37 °C to an OD600 of 0.2–0.25 in BHI supplemented sucrose (500 mM). Ampicillin (10 μg ml⁻¹) was added and cultures were further grown for 2 h. Cultures were centrifuged at $5000 \times g$ for 10 min at 4 °C and washed twice with ice-cold sucrose–glycerol washing buffer (SGWB; 10% glycerol, 500 mM sucrose). Lysozyme from chicken egg white (Sigma) was added to a final concentration of 10 μg ml⁻¹ and incubated at 37 °C for 20 min. Cultures were centrifuged at $3000 \times g$ for 10 min and washed in SGWB twice and finally resuspended in SGWB, aliquots were made and stored at −80 °C until used. The electrocompetent *L. monocytogenes* WT strain was separately transformed with pMAD::*rsbR1* (T209A), pMAD::*rsbR1* (T241A). Electroporated cells were plated in BHI supplemented with erythromycin. Chromosomal integration and subsequent excision was achieved through a two-step recombination as previously described[36]—the obtained transformants *L. monocytogenes* colonies were inoculated in BHI broth supplemented with erythromycin and grown at non-permissive temperatures of 40 °C for 24 h. Cultures were diluted and plated in BHI agar supplemented with erythromycin and grown for 24 h at 40 °C. the obtained integrant colonies were inoculated in BHI and grown at permissive temperatures of 30 °C. Dilutions of 1:100 were made in fresh BHI every 12 h. Simultaneously, serial dilutions were made at each passage and plated in BHI agar. Loss of the mutagenic plasmid was assessed by streaking the same colony in both BHI and BHI supplemented with erythromycin. cPCR with the primers *rsbR1*_upflank_F paired with either *rsbR1* (T209A)_R or *rsbR1* (T241A)_R, were used to identify the chromosomal mutation in the respective genes *rsbR*.

**RNA extraction and RT-qPCR**. *Cultures* of *L. monocytogenes* EGD-e and its isogenic mutants were grown until mid-log phase. Cultures were diluted in RNAlater™ (Sigma) at a 1:5 ratio to stop the transcription. Total RNA was extracted using an RNeasy minikit (Qiagen) according to the manufacturer's recommendations. Cells were disrupted by bead beating twice in FastPrep-24 at a speed of 6 m s⁻¹ for 40 s. DNA was removed with Turbo DNA-free (Invitrogen) according to the manufacturer's recommendations. The RNA integrity was verified by electrophoresis in 0.7% agarose gels. SuperScript™ III First-Strand Synthesis System (Invitrogen) was used to synthetize cDNA according to the manufacturer's recommendations. cDNA was quantified using NanoDrop 2000c (Thermo Scientific) and diluted to a final concentration of 7 ng ml⁻¹. RT-qPCR was performed

using a Quanti-Tect SYBR green PCR kit (Qiagen) and primers for the target genes (Supplementary Table 2). Primer for the target genes 16 S, *lmo2230* and *inlA* were previously tested using cDNA. Samples were analyzed on LightCycler 480 system (Roche) with the following parameters: 95 °C for 15 min; 45 cycles of 15 s at 95 °C, 15 s at 53 °C, and 30 s at 72 °C; a melting curve drawn for 5 s at 95 °C and 1 min at 55 °C, followed by increases of 0.11 °C s⁻¹ until 95 °C was reached; and cooling for 30 s at 40 °C. Cycle quantification values were calculated by using LightCycler 480 software version 1.5.1 (Roche) and the Pfaffl relative expression formula[37,38]. The 16S rRNA gene expression was used as a reference gene. Results are expressed as Log₂ relative expression ratios normalized against the expression of *L. monocytogenes* WT strain in the absence of stress. Three independent biological replicates were performed.

**Whole-genome sequencing and single nucleotide polymorphism (SNP) analysis**. The gDNA of all mutants' strains constructed in this study was extracted using DNeasy Blood and Tissue kit (Qiagen) according to the manufacturer recommendations. The obtained genomic material was analyzed via Illumina sequencing by MicrobesNG (Birmingham, UK). The resulting trimmed reads were analyzed using Breseq software[39] to identify additional SNP in the mutant strains chromosome. The nucleotide sequence of *L. monocytogenes* EGD-e chromosome (NCBI RefSeq accession no. NC_003210.1) was used as reference in this analysis.

**Homology modeling of RsbR paralogs in *L. innocua***. The sequences of the paralogs were detected in the *L. innocua* Clip11262 genome using the DELTA-BLAST algorithm with RsbR as the query. The homology models of the paralogs were then built using the default modeling script of Modeller (version 9.24)[40], using the model of RsbR as template. For each sequence, 20 models were generated and picked based on their Z-DOPE score. The structures were manually inspected for clashes and errors using the Chimera and Coot software packages.

**Mass spectrometry**. Mass spectrometric analysis of SDS-PAGE separated proteins was carried out as follows: protein bands were washed with 50 mM NH₄HCO₃, 50 mM NH₄HCO₃/acetonitrile (3/1), 50 mM NH₄HCO₃/acetonitrile (1/1) and lyophilized. After a reduction/alkylation treatment and additional washing steps, proteins were *in gel* digested with trypsin (Trypsin Gold, mass spectrometry grade, Promega) overnight at 37 °C. The resulting peptides were sequentially extracted with 50 mM NH₄HCO₃ and 50 mM NH₄HCO₃ in 50% acetonitrile. After lyophilization, peptides were reconstituted in 20 μl 1% TFA and separated by reversed-phase chromatography. An UltiMate 3000 RSLCnano System (Thermo Fisher Scientific, Dreieich) equipped with a C18 Acclaim Pepmap100 pre-concentration column (100 μm i.d. ×20 mm, Thermo Fisher Scientific) and an Acclaim Pepmap100 C18 nano column (75 μm i.d. ×250 mm, Thermo Fisher Scientific) was operated at flow rate of 300 nl/min and a 90 min linear gradient of 4–40% acetonitrile in 0.1% formic acid. The LC was online-coupled to a maXis plus UHR-QTOF System (Bruker Daltonics, Bremen) via a CaptiveSpray nanoflow electrospray source. Acquisition of MS/MS spectra after CID fragmentation was performed in data-dependent mode at a resolution of 60,000. The precursor scan rate was 2 Hz processing a mass range between $m/z$ 175 and $m/z$ 2000. A dynamic method with a fixed cycle time of 3 s was applied via the Compass 1.7 acquisition and processing software (Bruker Daltonics). Prior to database searching with Protein Scape 3.1.3 (Bruker Daltonics) connected to Mascot 2.5.1 (Matrix Science), raw data were processed in Data Analysis 4.2 (Bruker Daltonics). Search parameters for searching the UniProt *Listeria innocua serovar 6a* database were as follows: enzyme specificity trypsin with 2 missed cleavages allowed, precursor tolerance 10 ppm, MS/MS tolerance 0.04 Da, variable modifications: deamidation of asparagine and glutamine, oxidation of methionine, carbamidomethylation or propionamide modification of cysteine, phosphorylation of serine, threonine, and tyrosine.

Phosphopeptide fragment spectra with a Mascot peptide ion-score of at least 20 were evaluated manually.

**Phosphorylation of RsbRS by RsbT**. Buffers of purified proteins were exchanged to kinase assay buffer (50 mM Tris–HCl pH 7.5, 50 mM KCl, 10 mM MgCl₂, 0.1 mM EDTA pH 8.0, 1 mM ATP) using centrifugal filtration units (Amicon) of 100 and 10 kDa MWCO for RsbRS and RsbT, respectively. Purified RsbRS and RsbT were incubated at 37 °C for 5 min and mixed at a ratio of 5 to 1 (w/w), to start the reaction resulting in approximately 20 RsbT molecules per RsbRS complex. Reaction was carried out at 37 °C and quenched at different time intervals by mixing with 6× Laemmli sample buffer and heating to 95 °C for 2 min. The samples were the run on a 17.5% SDS–PAGE gel and stained with the Pro-Q™ Diamond Phosphoprotein Gel Stain (Thermo Fisher Scientific) according to the manufacturer's instructions. The gels were imaged on a GelDoc Go Gel Imaging System (Bio-Rad) using the settings for Ethidium Bromide gel imaging. Gels were quantified by measuring the intensity of the bands in Fiji[41], subtracting the measured background, and adjusting for total measured signal intensity between gels in Microsoft Excel.

**Reporting summary**. Further information on research design is available in the Nature Research Reporting Summary linked to this article.

## Data availability

The atomic model of the complex presented in this manuscript has been deposited in the wwPDB with the ID 7B0U. The corresponding EM map can be found in the EMDB under the code EMD-11971. The data for Fig. 3 can be found in the Supplementary Data 1 file supplied with the manuscript. All other data available upon reasonable request.

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

## Acknowledgements

This project has received funding from the European Union's Horizon 2020 research and innovation program under the Marie Skłodowska-Curie grant agreement No. 721456. Cryo-EM was carried out in the cryo-EM facility of the Julius-Maximilian University Würzburg. A.M. also thanks Maxi Kunzmann for help with the ProQ diamond gels during her internship.

## Author contributions

A.M. purified all protein samples used in the study and performed the biochemical assays, L.F. analyzed the cryo-EM data, M.G.M. built and analyzed the structure models, D.N.G. performed the *Listeria* in vivo experiments, S.K. cloned the *Li*RS and *Li*T proteins, M.C. and S.E. did bioinformatical analysis, and A.B. did the MS analysis. A.M., L.F., C.W., J.M.-W., R.J.L., C.O.B., J.P.-F., and C.Z. interpreted the data and wrote the manuscript.

## Funding

## Competing interests

The authors declare no competing interests.
