## [Transparent Peer Review File · Communications Biology]

Reviewers' comments:

Reviewer #1 (Remarks to the Author):

This manuscript describes the molecular mechanism of a bacterial stress response via stressosome. The stressosome is a stress-signaling supramolecular complex consisting of two-types of STAS proteins and one type of kinase. The environmental stress level will affect phosphorylation states of the STASs, but the mechanism of its allosteric regulation is unclear until now. To investigate the mechanism, there were so many efforts including stressosome structures from *Bacillus subtilis* and *Listeria monocytogenes* determined using CryoEM. This report shows new high resolution structure from *Listeria innocua*, which revealed the structure of unexpectedly phosphorylated RsbR-T241 in the complex. This depicts the possible switching of alpha 3 helix structure and its effect to RsbT releasing and sigma-B activation. The experiments and data handling described here are technically sound, and the manuscript is worthy of publication. However, there are two minor concerns described below.

1. TPO241-Thr209 interaction in RsbR: The author concluded the importance of the interaction between two residues via phosphorylation to release RsbT from RsbS. T209A mutation showed same effect of T241A mutation and it might reduced interaction to TPO241. However, there is a small concern on the T241A mutation, which affects the kinase activity to T209 by RsbT.
2. Recovery from the stressed condition: I would like to ask you to discuss whether the proposed model consistent with steps in recovery from the stressed condition or not. RsbS S59 (in *B. subtilis*) will be dephosphorylated first, because RsbX might easily access there similarly to RsbT binding without steric hindrance of RsbR turrets. However, there is phosphorylated RsbR T209 and then it does not introduce RsbT there.

Reviewer #2 (Remarks to the Author):

In this manuscript, the authors determined the cryoEM structure of *Listeria innocua* stressosome consisting of RsbR and RsbS. The resolution limits were 3.87 Å for D2 symmetry showing turrets (RsbR N-terminal domain) and 3.45 Å for icosahedral symmetry. In the structure, the authors found different conformations of RsbR N-terminal domain and suggested that the phosphorylation of T241 in RsbR could induce the conformational changes. In addition, they proposed a stressosome binding manner of RsbT based on homology modeling.

Major comments

1. Currently, some cryoEM structures of RsbR/RsbS complexes have been determined. Though authors found T241 phosphorylation regulates SigB signaling pathway, it is insufficient to explain allosteric activation of stressosomes.
2. Authors proposed a stressosome binding mode of RsbT based on homology modeling (Fig 5 and 6). Because RsbT binding sites on the stressosome were not determined clearly, the homology modeling is not adequate to explain the RsbT binding mode without any experimental results. To elucidate the RsbT binding mode, the authors need to determine the cryoEM structure of the stressosome holo-complex consisting of RseR, RsbS and RsbT.
3. Density maps are missing in Fig 3 and 4. Showing density maps with the models would be better to make sure whether it is an experimental structure-based model.
4. The statistics and methods of the cryoEM structure refinement are insufficient in supplementary table 1 and methods.
5. The result section includes details of the methods that should be placed in the method section.

Minor comments

1. In line 41, the entire complex needs to be modified to RsbRA/RsbS complex. The entire complex consists of three subunits (RsbR, RsbS and RsbT).
2. In line 172, "Thr241 in LiRsbR and the corresponding Thr88 in LiRsbS are conserved in the STAS

domains of RsbR and RsbS of different *Listeria* species as well as in *Bacillus subtilis*." – It would be better if the authors show sequence alignment results.

3. In line 181, during exponential growth in a complex broth medium (fig. 3g, h). there is no such figure (3g and 3h) in this manuscript.

4. Fig. 5 (a-d) does not match the text description (Fig5 a-f).

5. In line 182, a similar bridged interaction, Thr209-PO3-Thr241, would be conserved by Thr205-PO3-Thr237 in *B. subtilis*. On what basis author can state this point.

6. In line 206 LiRsbS/S  LiRsbR/S

Reviewer #3 (Remarks to the Author):

The manuscript titled "Molecular insights into an allosteric activation mechanism for the stressosome" tried to explain the conformational changes in the STAS domain linked to phosphorylation in RsbR protein using a 3.45Å structure of STAS and 3.87Å structure of the entire complex. With the help of docking studies, the authors propose a mechanism of stress-induced conformational changes in the N-terminal of RsbR. Overall the cryo-EM structure quality seems good, and the authors tried to explain the allosteric role by observing the structural changes. However, there are few issues with the manuscript, as listed below.

Major issues

1. There are already two cryo-EM stressosome structures published at about similar resolution from different organisms. Therefore, this manuscript does not offer much new information, and authors have not compared all those structures in detail. If there is no word-length issue, some comparison data can be moved to the main manuscript from supplementary data.

2. The authors tried to extrapolate structural changes to allosteric regulation with existing literature. Further validation experiments could have added strength to the manuscript.

3. This paper has not elaborated the role of the found allosteric site and the importance of amino acid residues for any biochemical studies. The sequence conservation and importance of amino acids present at the allosteric site could have been included. Including these details will enhance the readability of the manuscript and give a better overall contextual body.

Rebuttal letter **Point-by-Point response**

We sincerely thank all three reviewers for their constructive and helpful comments. They helped to improve the manuscript. We were able to change the manuscript according to most of the concerns raised. Changes in the revised manuscript are shown in red. We have also added Supplementary Figures as requested to strengthen our line of argumentation. In this rebuttal letter we have listed point-by-point our responses to all concerns raised and commented how we changed the manuscript accordingly. For better understanding, the reviewers' comment is always in blue, while our answers are in black and italic, exchanged text in the manuscript is in black.

Reviewer #1

1. TPO241-Thr209 interaction in RsbR: The author concluded the importance of the interaction between two residues via phosphorylation to release RsbT from RsbS. T209A mutation showed same effect of T241A mutation, and it might reduce interaction to TPO241. However, there is a small concern on the T241A mutation, which affects the kinase activity to T209 by RsbT.

*This is a valid concern as any mutation might interfere even indirectly with the phosphorylation process. However, from a structural point of view such a scenario is not very likely as T241A is quite distant from the catalytic side of RsbT when bound to RsbR. In addition, we can refer here to new data from the co-authors **Duarte Guerreiro** and **Conor O'Byrne**. They have finalized a separate manuscript showing that T241A is still phosphorylated in *L. monocytogenes*, which will be submitted within the next days. These data confirm that the alanine mutation does not affect the kinase activity of RsbT towards RsbR.*

2. Recovery from the stressed condition: I would like to ask you to discuss whether the proposed model consistent with steps in recovery from the stressed condition or not. RsbS S59 (in *B. subtilis*) will be dephosphorylated first, because RsbX might easily access there similarly to RsbT binding without steric hindrance of RsbR turrets. However, there is phosphorylated RsbR T209 and then it does not introduce RsbT there.

This is indeed an important aspect brought up by this reviewer. In the recovery of the stress condition, it is most likely that RsbT would dissociate from RsbS first, allowing the access of RsbX to dephosphorylate RsbS (see figure below);

*On one hand, a bound RsbT to the adjacent RsbR would not hinder RsbS-RsbX interaction. On the other hand, a bound RsbT to RsbR is incompatible with the binding of RsbX to RsbR. The dephosphorylation of RsbS may occur before RsbT is released from RsbR. From a structural point of view dephosphorylation of RsbS releases the tension on α 2-helix of the neighboring RsbR connected to RsbS by interactions shown in **fig. 4 f** of the manuscript. This interaction is likely involved in the stabilization of the α 2-helix of RsbR in a position bracing the interaction of phosphorylated T209 with T241.*

In summary, the dephosphorylation of the S56 in RsbS would support the conformational transition of α 3-helix in RsbR from the “up-“ to the “down-state”. In further events, RsbX would dephosphorylate RsbR, resetting the stressosome to process imminent stress.

We have added the following paragraph addressing this specific point in the discussion of the revised manuscript on page 13, first paragraph:

“Our model can be extended to the recovery following a stress activation event. We assume that RsbT would dissociate from RsbS first, allowing the access of RsbX to dephosphorylate RsbS. A still bound RsbT to the adjacent RsbR would not hinder such a RsbS-RsbX interaction and dephosphorylation of RsbS may occur before RsbT is released from RsbR. The dephosphorylation of RsbS releases the tension on α 2-helix of the neighbouring RsbR connected to RsbS (**fig. 4 f**). Dephosphorylation of S56 in RsbS would support the conformational transition of α 3-helix in RsbR from the “up“ to the “down-state”. In further events, RsbX would dephosphorylate RsbR, resetting the stressosome to process imminent stress.”

Reviewer #2:

Major comments

1. Currently, some cryoEM structures of RsbR/RsbS complexes have been determined. Though authors found T241 phosphorylation regulates SigB signaling pathway, it is insufficient to explain allosteric activation of stressosomes.

We acknowledge the point of the reviewer that the allosteric mechanism we propose is based on the interpretation of the docking data to the structure. Nevertheless, we would like to point out that our new structure shows a lot of more detail than the structures published so far, especially with respect to conformational changes. We have found T241 phosphorylated, but also observed significant secondary structure conformational changes, which explain not only the very different RsbT interaction between R and S, but also suggest for the first time a structural rationale for a RsbT-RsbT interaction.

A similar comment was also raised by reviewer #3 which made us understand that we have not clearly mentioned this in the manuscript. We have added in the beginning of the discussion (page 10) a statement to highlight the originality of our findings in comparison to the already published stressosome structures:

*“The LiRsbRS complex presented in this manuscript provides a unique opportunity to observe conformational changes in response to differences in phosphorylation especially in RsbR. Different to the recently published structure of the *Listeria monocytogenes* complex²¹, we were able to resolve helix 3 and its significant conformational changes by using thorough classification and reconstruction procedures that prevented an artificial averaging over orientations when applying the D2 symmetry. The high quality of the D2 map with a completely resolved STAS domain and linker helix allow now for first insights on how the stress signal is translated into a conformational change upon activation.”*

*However, we agree that a conformational change in one isolated structure is not a mechanism. Therefore, we have changed our title to “**Molecular insights into intra-complex signal transmission during stressosome activation**”. As our model provides molecular insight how the stressosome exploits the trimeric architecture of the functional RsbR-RsbS-RsbR unit in response to a stress signal we feel that this title summarizes our findings better than the previous title.*

2. Authors proposed a stressosome binding mode of RsbT based on homology modeling (Fig 5 and 6). Because RsbT binding sites on the stressosome were not determined clearly, the homology modeling is not adequate to explain the RsbT binding mode without any experimental results. To elucidate the RsbT binding mode, the authors need to determine the cryoEM structure of the stressosome holo-complex consisting of RsbR, RsbS and RsbT.

*We agree that an RsbR-RsbS-RsbT complex structure is important, and we have performed several attempts to determine its structure. However, others and we have not yet succeeded to obtain the RST complex stable in vitro. We would like to refer to the article of Williams et al on the structure of the *L. monocytogenes* stressosome, where the authors state that the RsbT kinase was added to the complex, but the density for the kinase was poorly resolved.*

*But we see the point of the reviewer and therefore we have added our attempts for a ternary complex to the manuscript (Result section: RsbT binding and release in LiRsbRS: page 8) and showed the 2D classes obtained from the RST complex in a supplementary **fig S11c and d**:*

*“We performed an additional size exclusion chromatography run with the LiRsbRS complex incubated with RsbT (**fig. S11c**). Although we obtained a ternary complex the amount of RsbT was varying between different experiments reflecting a highly dynamic interaction in vitro like what was observed for *LmRsbRST* binding²¹. A negative stain analysis of the co-*

eluted ternary complex suggested additional density for RsbT (**fig. S11 d**), however, we were not able to obtain a stable LiRsbRST complex suitable for high resolution structure determination.”

*Moreover, we have strengthened our statement that the binding mode seen in the crystal structure of SpoIIAA-SpoIIAB is a valid way to determine the binding poses of RsbT, owing to the high structural conservation of the STAS domains and their phosphorylation sites and of the kinase. The recently released crystal structures of RsbV-RsbW also confirm the same binding mode (**Structural insights into the regulation of SigB activity by RsbV and RsbW**-doi10.1107/S2052252520007617). We have referred now also to this structure when describing the docking experiments in the result section page:*

“To investigate the interaction of RsbT with the RsbRS complex we performed a docking study using two structurally well described STAS domain complexes, the crystal structures of SpoIIAB-SpoIIAA (pdb entry code 1TIL and 1TH8) and of RsbV-RsbW (pdb entry code 6M37)⁴¹, as blueprint.”

3. Density maps are missing in Fig 3 and 4. Showing density maps with the models would be better to make sure whether it is an experimental structure-based model.

*We have added a supplementary figure of the structure with the densities - **fig S10**. We decided that density in Fig. 3 and Fig.4 would probably hamper their readability and make them much harder to understand.*

4. The statistics and methods of the cryoEM structure refinement are insufficient in supplementary table 1 and methods.

We have updated the table and methods section with the requested information in Table 1 on page 21 of the Supplementary information and expanded the methods section regarding model building. We also added text regarding the processing of the cryoEM dataset.

“For data pre-processing, MotionCor2 and CTFFIND4 were used^{30,31} Afterwards, 105,020 particles were picked, extracted and classified in RELION 3.0³². After 6 rounds of 2D classification, a total of 53,487 particles were selected for further processing. An initial model was reconstructed using RELION 3.0, and all the particles were first subjected to global angular search three-dimensional (3D) classification in 60 iterations with four classes and step size of 7.5°, then the particles from the best class was subjected to the 2nd round of 3D classification with four classes and 3.75° local angular search step in 60 iterations. At this stage, 32,031 good particles were combined from the best 2 classes for the further 3D auto

refinement. The 3D refinement was done by the non-uniform refinement in Cryosparc V2²² using the corresponding map from 3D classification as the reference.”

5. The result section includes details of the methods that should be placed in the method section.

We have changed the text in the results section accordingly and moved some of the methods to the Methods section for easier reading.

Minor comments

1. In line 41, the entire complex needs to be modified to RsbRA/RsbS complex. The entire complex consists of three subunits (RsbR, RsbS and RsbT).

We agree that the “entire complex” is a term best used to describe the full RsbRST complex. We have changed our original wording as follows:

*“Here we present the map of *Listeria innocua* RsbR1/RsbS complex at resolutions of 3.45Å for the STAS domain core in icosahedral symmetry and of 3.87Å for the STAS domain and N-terminal sensors in D2 symmetry, respectively.”*

2. In line 172, “Thr241 in LiRsbR and the corresponding Thr88 in LiRsbS are conserved in the STAS domains of RsbR and RsbS of different *Listeria* species as well as in *Bacillus subtilis*.” – It would be better if the authors show sequence alignment results.

*We appreciate the comment. We added an alignment panel to **fig 3 e** in the revised manuscript and changed the text accordingly at the bottom of page 7 in the “Models of the LiRsbR and LiRsbS STAS domains” section.*

3. In line 181, during exponential growth in a complex broth medium (fig. 3g, h). there is no such figure (3g and 3h) in this manuscript

Thank you, the text has been changed to reflect the information of the figure.

4. Fig. 5 (a-d) does not match the text description (Fig5 a-f).

We have updated the text to match the figures as some panels were changed during editing, thank you for pointing this out. We updated the text in the last paragraph of “RsbT binding and release in LiRsbRS”, at the top of page 9, to match the current figure panels.

5. In line 182, a similar bridged interaction, Thr209-PO3-Thr241, would be conserved by Thr205-PO3-Thr237 in *B. subtilis*. On what basis author can state this point.

In the alignment we added to the revised manuscript (see comment 2) we show the conserved Thr241 in the STAS domain (fig 3e). Since the relative positions of the phosphorylation site T205 and T237 are the same on the STAS domain, we propose this is a conserved interaction.

6. In line 206 LiRsbS/S  LiRsbR/S

Thank you, was changed accordingly.

Reviewer #3:

1. There are already two cryo-EM stressosome structures published at about similar resolution from different organisms. Therefore, this manuscript does not offer much new information, and authors have not compared all those structures in detail. If there is no word-length issue, some comparison data can be moved to the main manuscript from supplementary data.

We agree that there are some stressosome structures out there, however, most of them have turret densities with very uneven density levels. We have specifically aligned our particles in a way to avoid artefacts caused by wrongful averaging of different orientations. Our map therefore provides a very high overall resolution also for the turret region and revealed distinct conformational changes (see also answer to comment 1 of Reviewer #2). We have added a section on the new findings of our structure in the beginning of the discussion. In addition, we added a more detailed comparison to other structures in the discussion, e.g., on the interfaces between STAS domains covered by Kwon et al. (2019) on page 11. We also provided important structural detail missing in Williams et al (2019), particularly the resolved helix 3. In summary, the better overall resolution allows for the most detailed view of interactions involved in stressosome activation.

2. The authors tried to extrapolate structural changes to allosteric regulation with existing literature. Further validation experiments could have added strength to the manuscript.

We agree with your statement, however, the significance of T241A in vivo is high in terms of SigB activation, as indicated by the upcoming publication from our colleagues (O'Byrne and Guerreiro).

3. This paper has not elaborated the role of the found allosteric site and the importance of amino acid residues for any biochemical studies. The sequence conservation and importance of amino acids present at the allosteric site could have been included. Including these details will enhance the readability of the manuscript and give a better overall contextual body.

We address the sequence conservation of T241A in a new panel in fig 3e. However, we have toned down the allostery as main outcome of our structural work and focused more on the conformational changes.

REVIEWERS' COMMENTS:

Reviewer #1 (Remarks to the Author):

The author clarified my concerns in the revised manuscript. So I think it is now suitable for publication in Communications Biology.

Reviewer #2 (Remarks to the Author):

The manuscript revision was not based on experimental results of the major concern. The density map in Fig S10 is not clear whether phosphorylation is properly assigned. Overall, no novelty to publish this journal appears to be found in this manuscript.

Reviewer #3 (Remarks to the Author):

Authors have addressed the concerns raised by us.

Rebuttal to comment of reviewer #2

We thank reviewer #2 for critically reading of our manuscript. Apparently, reviewer #2 was not convinced from the density maps we added to the revision. Therefore, we now added a large zoom-in into the density of TPO 241 in supplementary figure 10, which unambiguously show the phosphorylation at this site. We cannot agree with reviewer #2 that our mechanism is not based on experimental data and would like to summarize again our findings.

The phosphorylation site was modelled, and both the density map and the model were accepted at the respective data banks meeting their quality controls.

In the density map the phosphate group at the position TPO 241 is clearly visible and hardly to miss interacting with T209 at a distance of 2.7Å.

Moreover, we presented mass spec data that unambiguously validate phosphorylation at this position, while no phosphorylation was detected at T209.

The manuscript contains functional measurements on the impact of T241 in

the *Listeria* stressosome, which strengthen our mechanistic interpretations on the conformational role of T209-TPO-T241, which is a new finding based on experimental data (structure, mass spec, mutagenesis study).

Finally, we could show in our high-resolution density map a conformational change in the RsbR Stas domain as response to phosphorylation in a region which was largely misinterpreted and modelled with a frame shift by a previous publication on the *Listeria monocytogenes* stressosome. In fact, the badly resolved region of this previously published *L. monocytogenes* map supports our findings that this is a flexible part of the stressosome. We would like to point out that we have processed our data in a way to avoid any averaging artefacts, which is often a problem in stressosome structure determination reflected by decreased or even missing densities in the sensory domains. Our careful processing with respect to the correct location of the D2 symmetry axis is one of the reasons why we were able to detect the conformational heterogeneity in the *Listeria* RsbR Stas domain, which we could trace back to phosphorylation.

We provided a straightforward explanation due to these conformational changes why the triangular arrangement of the RsbR-RsbS-RsbR functional unit is important for stress signal transmission. We performed docking experiments with RsbT to our structure, which pointed to an intriguing RsbT-RsbT interaction. All these data and the proposed mechanism is based on experimental data.

In summary we hope we can finally convince reviewer #2 that our structure function study on the *Listeria innocua* stressosome comprise structural mechanistic insights which were not present to date in other stressosome structures.